# Puriton attenuates the asthma severity in ovalbumin-induced murine model via balancing Th1/Th2 and inhibiting inflammation

So-Hyeon Bok[1☉], Hae Eun Jang[2,3☉], Kwang-Ho Kim[4], Dae-Hun Park 🄳[1]*

**1** College of Korean Medicine, Dongshin University, Naju, Jeonnam, Korea, **2** College of Veterinary Medicine, Seoul National University, Seoul, Korea, **3** ORIENT Inc. Seungnam, Kyunggi, Korea, **4** Kadesh, Inc., Garden Grove, California, United States of America

☉ These authors equally contributed in this study.
* dhj1221@hanmail.net

## Abstract

In 2019, 262 million asthma patients were estimated, with 455 thousand deaths caused by asthma. It is an incurable chronic inflammatory respiratory disease and is more severe in the elderly and in the young. Forty BALB/c mice were divided into 5 groups of eight mice each: vehicle group (CON), asthma group (OVA), positive drug group (DEX), and Puriton (700 and 1400 µL/head/day). After animal experiment all mice were narcotized for collecting bronchoalveolar lavage fluid (BALF) and blood and then anesthetized for sampling the lungs. White blood cell (WBC) and differential count in BALF and immunoglobulin E (IgE) in serum were measured. Lung tissues were used for histopathological and immunohistopathological studies. Treatment with Puriton decreased the populations of WBC and neutrophil and the level of IgE. It prevented OVA-induced morphological changes in the lung and increased the expression levels of T helper 2 (Th2) cell-related cytokines such as interleukin- (IL-)4, IL-5, and IL-13. It inhibited inflammatory cytokines such as tumor necrosis factor alpha (TNF-α) and IL-6. The nuclear factor kappa-light-chain-enhancer of activated B cells (NF-κB)/ cyclooxygenase-2 (COX-2)/ prostaglandin E2 (PGE$_2$) pathway is a significant inflammatory pathway. Treatment of the subjects with Puriton resulted in the inhibition of the expression of phosphorylated (*p*)-NF-κB, COX-2, and PGE$_2$ in both the nucleus and cytoplasm. From the results we concluded that Puriton is a promising drug candidate as asthma treatment.

## Introduction

Asthma is a chronic inflammatory hyperresponsiveness in respiratory system which is caused by indoor allergens such as house dust mite, cockroaches, and pet dander and outdoor allergens such as mice, pollen, peanut protein, and mold [1–3]. Symptoms of asthma include wheeze, shortness of breath, chest tightness, and cough

**Data availability statement:** All relevant data are within the manuscript.

**Funding:** The author(s) received no specific funding for this work.

**Competing interests:** I have read the journal's policy and the authors of this manuscript have the following competing interests: Kwang-Ho Kim. None of the authors have any financial conflict of interest that might be construed to influence interpretation of this manuscript. This does not alter our adherence to PLOS ONE policies on sharing data and materials.

which are caused by airflow limitation [4]. Airway limitation results from the representative morphological changes in pulmonary system in asthma patients such as airway remodeling including epithelial cell hyperplasia, apoptosis, fibrosis [5], mucous hypersecretion [6], and inflammatory cell infiltration such as neutrophils (NE), M2 macrophages, and eosinophils (EO) [7,8].

In 2019, 262 million asthma patients were estimated, with 455 thousand deaths caused by asthma [9]. The incidence and prevalence of asthma are higher in the young but the morbidity and mortality of that are higher in adults, especially the old [10]. The polarized severity of health problems depending on the age should increase as time goes by because in the world in 2019, 9.1% of population was 65 years and older but by 2050, 16.7% will be over that age [11].

The maintenance of a balanced ratio between Th1 and Th2 cells is imperative for the stability of immune responses in bio-organisms. In certain allergic diseases, such as asthma and atopic dermatitis, Th2 cell-related factors have been observed to exert a dominant effect on both Th1 cells and regulatory T (Treg) cells [12]. In asthma patients, including Th2 cell transcription factor, GATA3 the high levels of Th2-cell related cytokines such as IL-4, IL-5 and IL-13 were observed [13]. Although one of Th1 cell's functions has been commonly known to modulate the activation and level of Th2 cell [14] from the recent study Th1 cell-related cytokine, interferon gamma (IFN-γ) had been reported to play keeping the inflammatory response in asthma with Th2 cell-related cytokines [12]. Regulatory T cell modulates the balance of Th1 and Th2 cells and Treg cell-related cytokines such as IL-6 and TNF-6 play an important role in asthma [15,16].

The management of inflammation in asthma patients is of paramount importance, and the NF-κB/COX-2/PGE$_2$ pathway represents the primary inflammatory pathway [17–19]. The activation of this pathway is induced by inflammatory cytokines such as IL-1β and TNF-α, and it plays a role in inducing inflammatory responses [20]. The enzyme phospholipase A2 is responsible for the induction of arachidonic acid within the cell membrane. The arachidonic acid that is generated is then converted into prostaglandins and leukotrienes by COX-2 and lipoxygenase, respectively [18].

As asthma is incurable disease current drugs are just used as symptom modifiers. Asthma drugs are categorized into: anti-inflammatory two groups such as anti-inflammatory drugs such as corticosteroid, leukotriene modifier and mast cell stabilizer and bronchodilators such as β-adrenergic drug, anticholinergic, and methylxanthine [21]. As most patients require an inhalator (nebulizer), it is difficult to administer treatment without one, and this is the therapeutic limit [22]. So many adverse effects of asthma drugs have been reported such as growth retardation, osteoporosis, immunosuppression, glaucoma/cataracts, and psychiatric problems [23–25].

Minerals micro-quantitatively exist in our body excluding calcium in bone but play various and important roles in maintaining our life such as gas exchange, fluid balance, and muscle contraction. Recently biological effects of them have been induced such as anti-cancer [26], anti-diabetes type 2 [27], antioxidative effect [28], bactericidal and virucidal efficacies [29], and cognitive function enhancement [30]. Puriton is composed of 20% biotite, 17% kaolinite, 16% montmorillonite, 12% serpentine,

8% mica, 4% feldspar, 4% vermiculite, 3% muscovite, 2% brucite, 2% limestone, 2% illite, 1% zeolite, 1% orthoclase, and 8% other minerals in sterilizing water at approximately pH 7.0 [29]. In our previous study we defined that Puriton controlled nine pathogens and opportunistic bacteria such as *Salmonella typhimurium, Escherichia coli, Pseudomonas aeruginosa, Alcaligenes faecalis, Staphylococcus aureus, Enterococcus faecalis, Micrococcus luteus, Mycobacterium smegmatis, and Bacillus subtilis* and two viruses such as Zika and Influenza A/Duck/MN/1525/81 under safety level [29].

In this study we investigated anti-asthmatic effect and therapeutic mechanism of Puriton.

## Materials and methods

### Ethics statement

Animal experiment was conducted after the approval of the Institutional Animal Care and Use Committee of Chonnam National University (CNU IACUC-YB-2018–36).

### Animal experiment

Animal experiment was conducted using 40 BALB/c female mice (Samtako Korea, Osan, Korea) and consisted of five groups: vehicle treated group for control (CON), asthma induction group (OVA), 1 mg/kg/day dexamethasone treated group as appositive control (DEX, SAMNAM Pharm, Chungchengnam-do, Korea, 02450) after asthma induction, and 2 dosage groups of Puriton (700 and 1400 μL/head/day, Kadesh, Inc., CA, USA, KP0352101) after asthma induction. For each group 8 heads were used that 4 animals were used for BALF analysis and 4 heads were used for the other experiments such as histopathological and immunohistochemical studies. Because of the possibility of some damage to lung tissue during the BALF collection process, four heads were used. The composition of Puriton was explained in our previous study [29]. To induce asthma forty animals were intraperitoneally injected with 20 μg of ovalbumin and 1 mg of aluminum hydroxide hydrate (Sigma-Aldrich, St. Louis, MO, USA, 77161) in 500 μl saline on days 1 and 8. From days 21–25, dexamethasone or Puriton was orally applied in the morning and in the afternoon, 5% ovalbumin was inhaled using nebulizer (3 mL/min, NE-U17, OMRON, Kyoto, Japan) for 30 min.

### BALF and serum IgE analysis

After the final treatment, all mice were anesthetized with 40 mg/kg Zoletil (Virbac, Fort Worth, TX. USA, BN 6NGW) and an adequate depth of anesthesia was assessed by tail pinch reflex. After collecting blood, they were euthanized by additional injections of Zoletil. BALF was collected from the trachea, and whole blood was collected from the heart using a flexible needle from four mice per group. The bronchoalveolar duct was washed with 0.4 mL of phosphate-buffered saline (PBS, Walkersville, MD, USA, 17-516F) and then, BALF was collected and subjected to centrifugation at 3,000 rpm for 5 min at 4°C using an M15R microcentrifuge (Hanil Scientific, Gyeonggi-do, Korea). The differential cell count of the harvested BALF was then performed using the Hemavet Multispecies Hematology System (Drew Scientific). To measure serum IgE levels, blood was collected and subsequently spun at 12,000 rpm for 15 min using an M15R microcentrifuge (Hanil Scientific). IgE levels were then measured using the OptEIA Mouse Kit (BD Biosciences, San Jose, CA, USA, 555248), and the results were acquired using a Multiskan SkyHigh device (Thermo Fisher Scientific, Waltham, MA, USA).

### Histopathological evaluation

The lungs were collected from mice that BALF was not collected from and fixed in 10% (v/v) formaldehyde (Daejung Chemicals & Metals, Gyeonggi, Korea, 4044–4400) for 1 month. After fixation, the samples were dehydrated using graded ethanol solutions from 99.9% to 70% and embedded in paraffin (Leica Biosystems, Richmond, IL, USA, 39601006). Paraffin embedded lung was longitudinally sectioned at 5 μm and stained with hematoxylin (Vector Laboratories, Newark, CA, USA, H-3401) and eosin (Merck Millipore, Darmstadt, Germany, HX57865344) to evaluate morphological changes.

To measure the level of mucous secretion in bronchioalveolar duct periodic acid (Sigma–Aldrich) and schiff stain (Merck Millipore, HX20146234) were conducted. The results were obtained using Zeiss AX10 Scope A1 microscope (Zeiss, Baden-Wurttemberg, Germany).

### Immunohistochemical analysis

To measure the change of immune balance-related helper T cell several cytokines such as Th1-, Th2-, and Th17-related cell cytokines and inflammation such as $PGE_2$ were immunohisochemically stained using ImmPRESS Universal Polymer Kit (Vector Laboratories, MP-7500). Several primary antibodies were used such as IFN-γ (Thermo Fisher Scientific, Waltham, MA, USA, MM700) and IL-12p40 (Santa Cruz Biotechnology, Santa Cruz, CA, USA, sc-57258) for Th1-related change, IL-4 (Santa Cruz Biotechnology, sc-73318), IL-5 (Biolegend, San Diego, CA, USA, 504302), and IL-13 (Santa Cruz Biotechnology, sc-1776) for Th2-related change, and TNF-α (BioVision, Milpitas, CA, USA, 3053R-100) and IL-6 (Novus Biologicals, Centennial, CO, USA, NBP1–77894) for Th17-related change and $PGE_2$ (Bioss, Woburn, MA, USA, bs-2639R). To quantify the result positive stained cells were counted in five non-overlapping fields of four independent immunostained lung sections in the group. Antigen retrieval was conducted and followed by the blocking serum to prevent nonspecific binding.

### Immunofluorescence analysis

The present study employed immunofluorescence analysis to evaluate the changes in the levels of p-NF-κB and COX-2 proteins. Lung tissue sections were subjected to a dehydration process using ethanol and then to antigen retrieval for a period of 40 min. The tissue samples were then incubated with primary antibodies diluted in 1% bovine serum albumin (BSA), including p-NF-κB (Invitrogen, MA5–15160) and COX-2 (Invitrogen, PA1–9032), for a duration of one hour. Secondary antibodies, including Alexa Fluor 488-conjugated anti-rabbit IgG (A3273; Invitrogen) and Alexa Fluor 555-conjugated anti-goat IgG (A32816; Invitrogen), were then conjugated. DAPI (62249; ThermoFisher Scientific, Waltham, MA, USA) was used for counterstaining. Images were acquired using a K1-Fluo confocal microscope (Nanoscope System, Daejeon, Korea).

### Statistical analysis

The results were represented as average ± standard deviation (SD). The difference of each group was measured by one-way analysis of variance (ANOVA) followed by Dunnett's multiple comparison test. Statistical significance was set at $p < 0.05$.

## Results

### Puriton controlled the OVA-induced increased in WBCs, NE, lymphocytes (LY) and IgE

Controlled the OVA-induced increased populations of WBCs, NE and LY in BALF. As shown in Fig 1A, compared to the level of WBC in CON OVA significantly increased the level of WBC to almost quadruple ($p < 0.001$) but DEX effectively suppressed that to one third ($p < 0.001$). Although the decrease level of that was lower by Puriton it dose-dependently down-regulated the level of WBCs ($p < 0.001$ or $p < 0.05$). Among components in WBC the populations of NE (Fig 1B) and LY (Fig 1C) were effectively suppressed by Puriton ($p < 0.001$ or $p < 0.05$). DEX effectively decreased the OVA-induced increased level of serum IgE and as a result in DEX group ($p < 0.05$). Puriton dose-dependently suppressed the level of IgE in blood (Fig 1D, $p < 0.05$).

### Puriton stopped asthma-related changes to the lungs

As shown in Fig 2A OVA induced the representative morphological changes in the lung such as mucous hypersecretion in bronchioalveolar duct, epithelial hyperplasia of bronchioalveolar duct, and inflammatory cell infiltration near vessels and bronchioalveolar ducts. Although a slight inflammatory cell infiltration was observed DEX effectively prevented the OVA-induced morphological changes in pulmonary system. In 700 μL/head/day Puriton treatment group the morphology of the

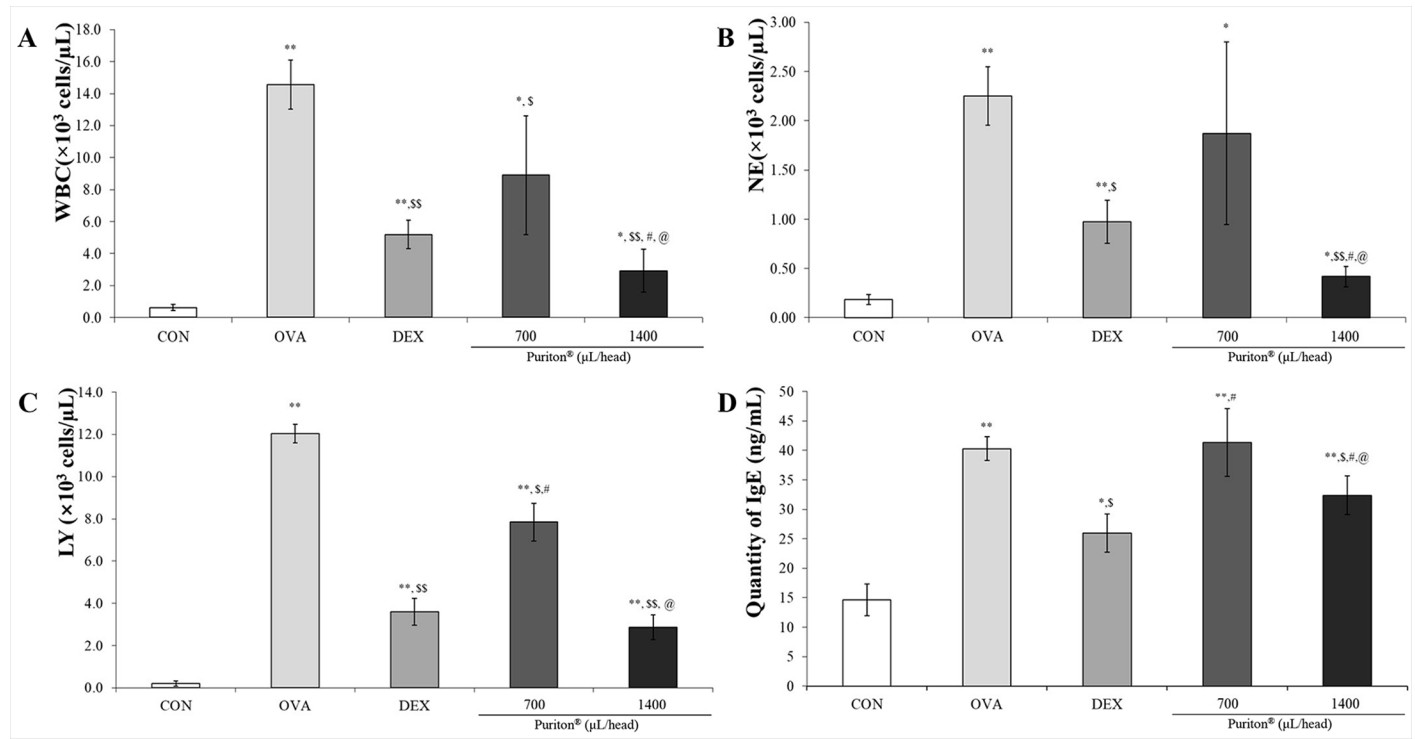

**Fig 1. Puriton's modulation effects on WBC composition in BALF and serum IgE.** (A) Puriton decreased the number of WBC which was increased by ovalbumin treatment. (B) Puriton suppressed neutrophil's (NE) population of WBC composition in BALF (C) Puriton effectively controlled lymphocyte's population of white blood cell composition in BALF. (D) Puriton dose-dependently suppressed the level of IgE in serum. The results were described by mean ± SD. N = 4. * $p < 0.05$ vs. CON, ** $p < 0.001$ vs. CON, $ $p < 0.05$ vs. OVA, $$ $p < 0.001$ vs. OVA, # $p < 0.05$ vs. DEX, ## $p < 0.001$ vs. DEX, @ $p < 0.05$ vs. Puriton 700 µL/head. CON; vehicle group, OVA; asthma group, DEX; positive drug group.

lung was like that in OVA group 1400 µl/head/day Puriton treatment significantly controlled the morphological changes by OVA. OVA increased the level of mucous to make mucous plug in the bronchioalveolar duct, but DEX decreased the level of that and Puriton dose-dependently suppressed the mucous secretion (Fig 2B).

### Puriton related Th1 cell-related cytokines such as IFN-γ and IL-12p40

As shown in Fig 3A, OVA promoted the release of IFN-γ protein from cells and the expression of that was significantly observed near bronchioalveolar duct and vessel ($p < 0.001$) but DEX controlled the expression of IFN-γ. Puriton appeared to decrease IFN-γ levels in the lung more than OVA. Like the result of IFN-γ the expression of IL-12p40 was increased by OVA treatment ($p < 0.001$) but was effectively controlled by DEX ($p < 0.05$). Puriton treatment down-regulated the expression of IL-12p40 (Fig 3B). Although the immunohistochemical analysis suggested that the control effect of Puriton on IFN-γ and IL-12p40 seemed effective based on quantitative evaluation, the effect of Puriton did not reach statistical significance (Fig 3C).

### Puriton dose-dependently suppressed Th2 cell-related cytokines

The expression of IL-4 protein in OVA treatment group significantly increased compared to CON but was inhibited by DEX treatment. Puriton dose-dependently suppressed the expression level of IL-4 (Fig 4A, $p < 0.001$). Puriton inhibited the increase of IL-5 level which was caused by OVA ($p < 0.001$) and especially 1400 µL/head/day Puriton suppressed the level of that like the result of DEX treatment (Fig 4B, $p < 0.001$). Like the results of IL-4 and IL-5 OVA significantly increased the level of IL-13 ($p < 0.001$) but DEX effectively suppressed that ($p < 0.001$). Puriton dose-dependently down-regulated the

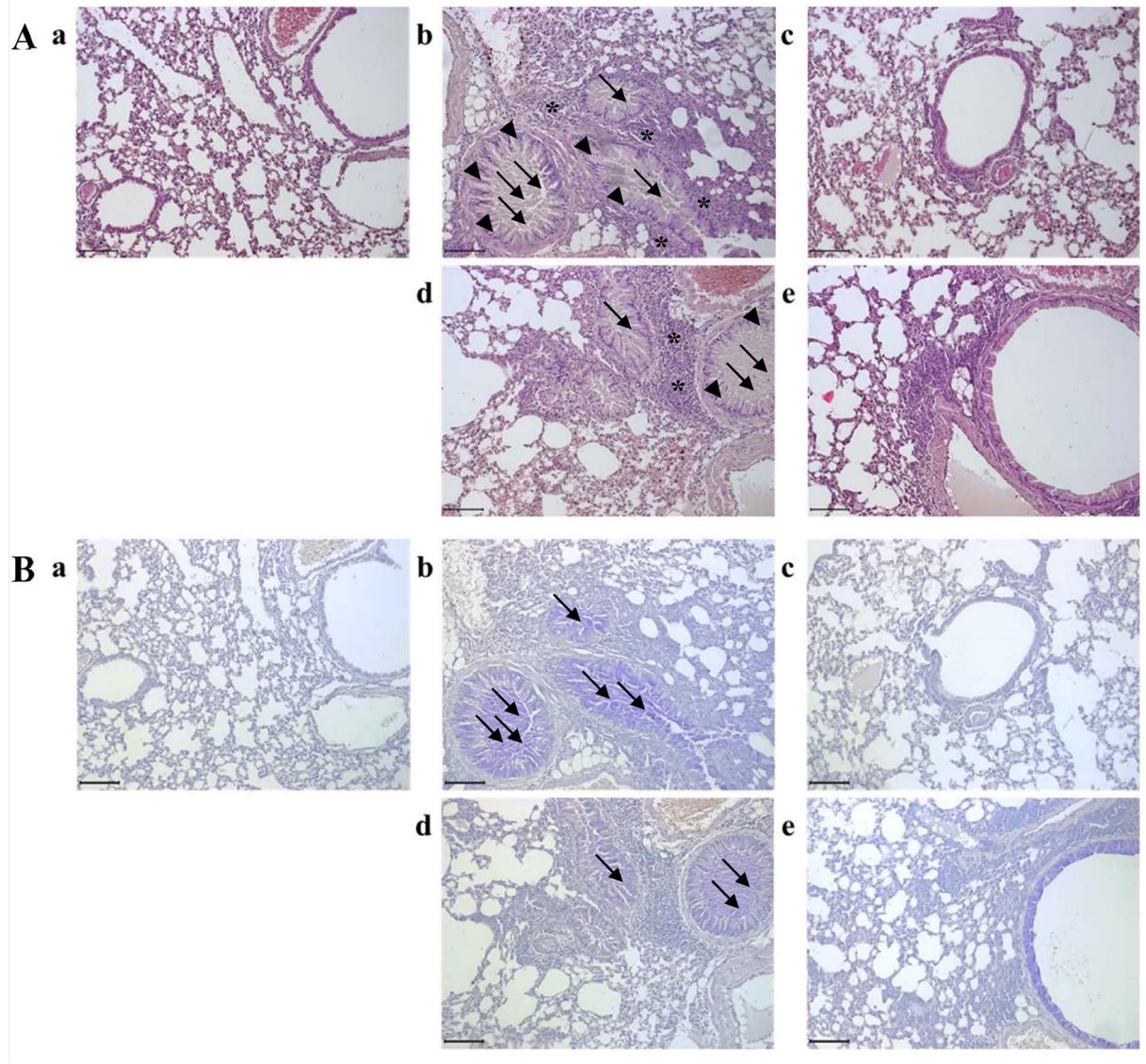

**Fig 2. Puriton's preventive effects on ovalbumin-induced histopathological changes in the pulmonary system.** (A) Puriton prevented representative histopathological changes such as mucous hypersecretion (arrow), epithelial cell hyperplasia (arrow head), and inflammatory cell infiltration (asterisk mark) in the lung. (B) Puriton dose-dependently suppressed ovalbumin-induced mucous secretion (arrow) in bronchioalveolar duct. a. CON, b. OVA, c. DEX, d. Puriton 700 µl/head, **e.** Puriton 1400 µL/head. Scale bar, 100 µm. Magnification, ×200.

level of IL-13 (Fig 4C, $p < 0.001$). As shown in Fig 4D, based on the quantitative analysis of Th2 cell-related cytokines in a dose-dependent manner Puriton suppressed the level of three cytokines such as IL-4, IL-5, and IL-13 and in particular inhibited the level of IL-5 like the level of that in DEX (Fig 4D, $p < 0.001$ or $p < 0.05$).

**Puriton dose-dependently down-regulated the levels of inflammation-related cytokines such as TNF-α and IL-6**

DEX decreased OVA-induced increased level of TNF-α ($p < 0.001$) and Puriton dose-dependently suppressed the releasing level of that (Fig 5A, $p < 0.05$). OVA induced the release of IL-6 ($p < 0.001$) but DEX effectively controlled that ($p < 0.05$).

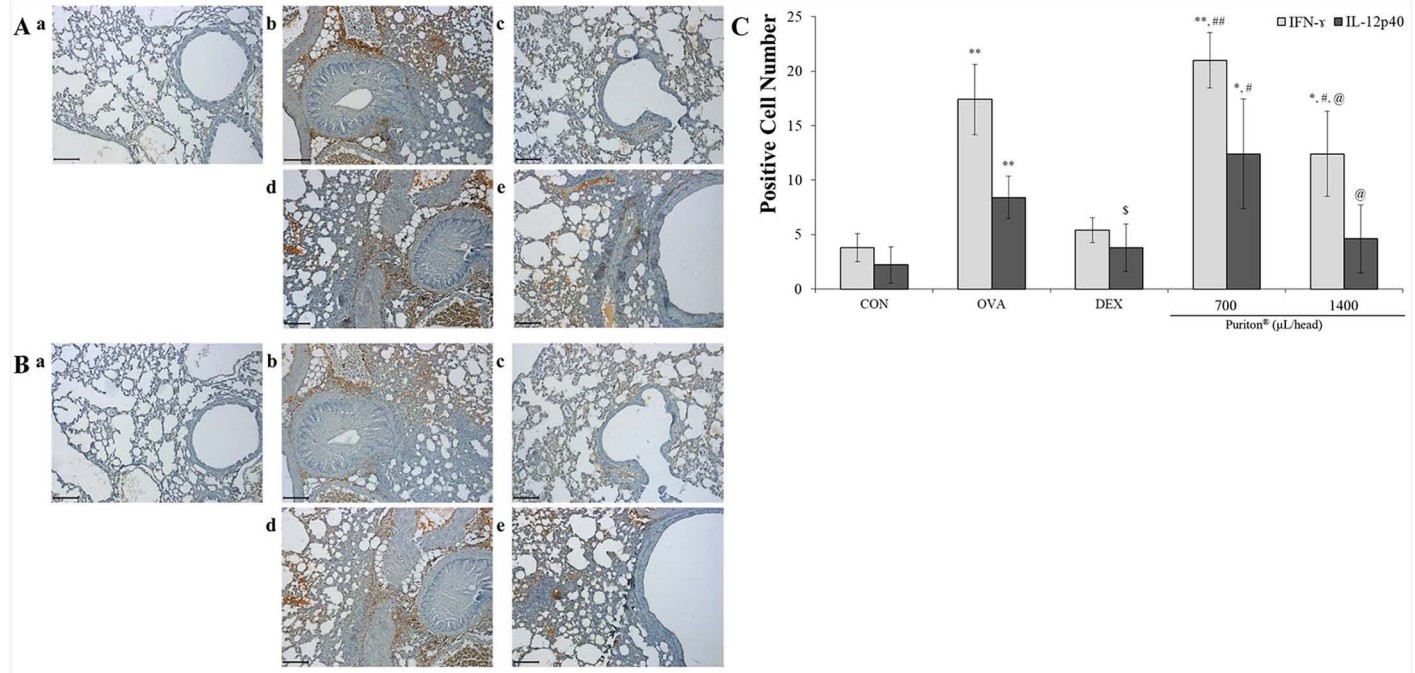

**Fig 3. Puriton had the tendency of decreasing expressions of Th1 cell-related cytokines such as IFN- γ and IL-12p40.** (A) Although there was no statistical significance Puriton inhibited the expression of IFN-γ protein. (B) Under no statistical significance Puriton had the tendency of suppressing ovalbumin-induced IL-12p40 release in the lung. a. CON, b. OVA, c. DEX, d. Puriton 700 μL/head, **e.** Puriton 1400 μL/head. Scale bar, 100 μm. Magnification, ×200. (C) The summary of anti-expression effects by Puriton on IFN-γ and IL-12p40. The results were described by mean±SD. * $p < 0.05$ vs. CON, ** $p < 0.001$ vs. CON, $ $p < 0.05$ vs. OVA, # $p < 0.05$ vs. DEX, ## $p < 0.001$ vs. DEX, @ $p < 0.05$ vs. Puriton 700 μL/head.

Puriton dose-dependently suppressed the level of IL-6 ($p < 0.001$ or $p < 0.05$) and especially in 1400 μL/head/day Puriton treatment group that was like the result in DEX (Fig 5B, $p < 0.001$). The summary of TNF-α and IL-6 levels demonstrated that Puriton exhibited a dose-dependent suppression of both cytokines ($p < 0.001$ or $p < 0.05$). Notably, treatment with 1400 μL/head/day of Puriton led to a significant downregulation of IL-6 levels ($p < 0.001$), comparable to the levels observed with DEX treatment (Fig 5C).

## Puriton inactivates the NF-κB/COX-2/PGE$_2$ pathway

Immunofluorescence revealed that Puriton led to a suppression of the expression of p-NF-κB (green fluorescence) and COX-2 (red fluorescence), which were increased by OVA treatment in the nucleus and cytoplasm. The administration of 1400 μL/head/day of Puriton resulted in a comparable outcome to that observed in the DEX treatment group (Fig 6A). Immunohistochemistry for PGE$_2$ expression showed that Puriton treatment reduced the PGE$_2$ (dark brown) levels increased by OVA treatment (Fig 6B). The administration of Puriton at a dosage of 1400 μL/head/day led to a reduction in PGE$_2$ levels to a level comparable to that observed in subjects treated with DEX (Fig 6C, $p < 0.001$).

## Discussion

When antigen presenting cells (APCs) meet allergen, APCs introduce it to Th2 cell. Th2 cell releases some cytokines such as IL-4, IL-5 and IL-13 and then these cytokines stimulate IgE activation. Although the half-life of IgE is short (< 1 day), its high affinity for mast and basophil allows easy interaction with allergens. After mast cell and/or basophil meets an allergen, leukotriene D$_4$ and PGE$_2$ are produced within 30 min, and within several hours, TNF-α and IL-4 are released in various situations such as asthma, allergic rhinitis, and atopic dermatitis [31–33].

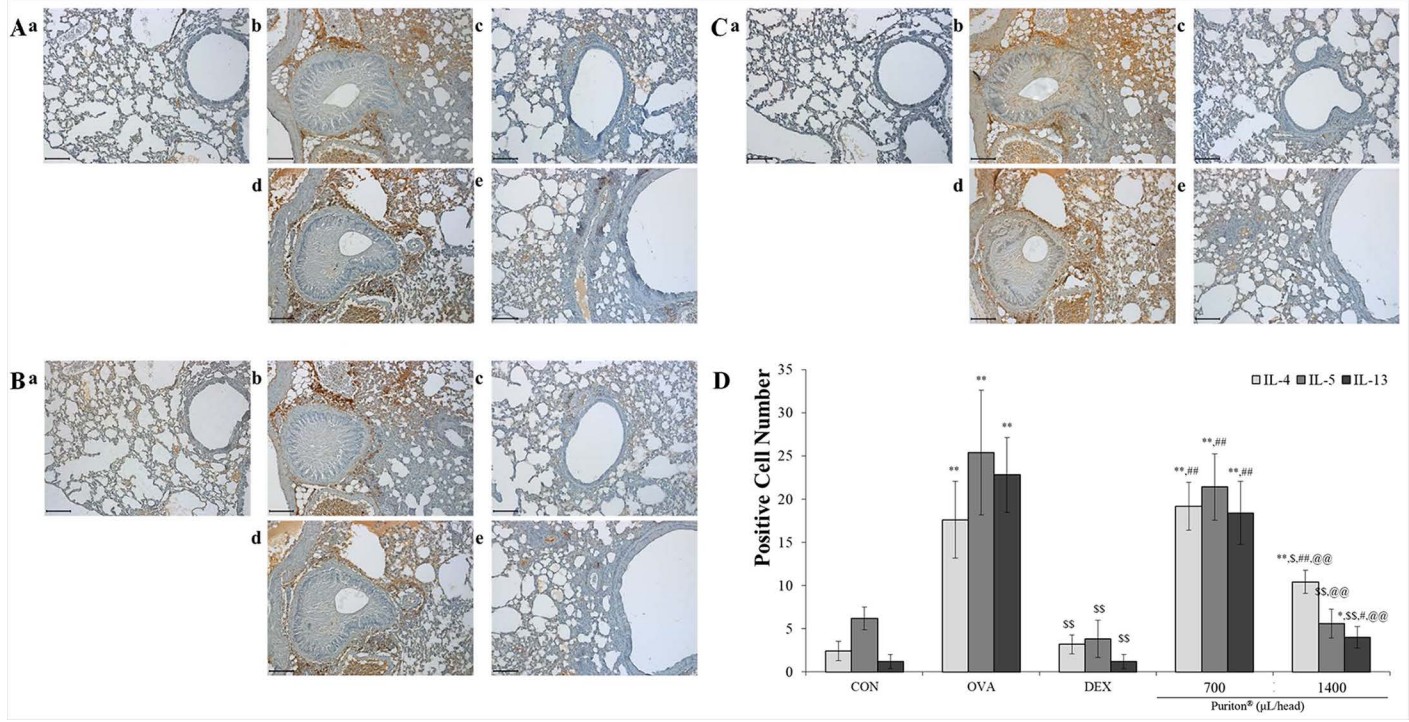

**Fig 4. Puriton dose-dependently inhibited the expression of Th2 cell-related cytokines such as IL-4, IL-5 and IL-13.** (A) Puriton dose-dependently down-regulated the release of IL-4 protein near the vessels and bronchioalveolar ducts. (B) Puriton almost completely inhibited the expression level of IL-5 protein which was caused by ovalbumin treatment. (C) Puriton effectively suppressed IL-13 expression. a. CON, b. OVA, c. DEX, d. Puriton 700 μL/head, e. Puriton 1400 μl/head. Scale bar, 100 μm. Magnification, ×200. (D) The summary of anti-expression effects by Puriton on IL-4, IL-5, and IL-13. The results were described by mean±SD. * $p<0.05$ vs. CON, ** $p<0.001$ vs. CON, $ $p<0.05$ vs. OVA, $$ $p<0.001$ vs. OVA, # $p<0.05$ vs. DEX, ## $p<0.001$ vs. DEX, @@ $p<0.001$ vs. Puriton 700 μL/head.

It has been hypothesised that an imbalance of Th1 and Th2 cells may be a contributing factor to the occurrence of asthma. Recent studies have provided substantial evidence that Th2 cell-related cytokines, such as IL-4, IL-5 and IL-13, play a pivotal role in the development and severity of asthma. In addition, Th1 cell-related cytokines, including IFN-γ and IL-12p40, have been shown to have a significant impact on asthma [12]. IL-4 released from allergen-bound Th2 cell stimulates the release of IL-4, IL-5, and IL-13, promotes IgE production from B cells, and accelerates inflammation in asthma patients [34]. In particular, IL-4 is suggested to play an important role in early phase of asthma occurrence via positive loop to produce Th2 cell-related cytokines and IgE, but IL-13 is considered to relate with the late phase of asthma, such as mucous hypersecretion and airway remodeling [35]. IL-5 concerns activations of B cell and eosinophil and to airway hyperreactivity [36,37]. IFN-γ plays double edged swords on asthma, as it not only suppresses recruitment of eosinophils and airway hyperresponsiveness but also stimulates recruitment of neutrophils and inflammation in the lung [38,39]. IFN-γ produces positive feedback Th1 cell-related cytokines, such as IFN-γ and IL-12, and suppresses the negative loop of Th2 cell-related cytokine, including IL-4 [40,41]. IL-12p40 is produced by neutrophil, dendritic cells, monocyte, and macrophage and is involved in some inflammatory responses, such as asthma and graft rejection [42]. Asthma is one of the chronic respiratory inflammatory diseases, and in almost asthma patients, the levels of inflammation-related cytokines, such as IL-6 and TNF-α, are elevated [31,32]. In particular, IL-6 and TNF-α are deeply associated with asthma severity, such as airway remodeling and fibrosis in the late stage [15,43]. We obtained results showing that Puriton effectively controlled the levels of some cytokines, such as IL-5, IL-6, and TNF-α (Fig 4B and 5). IL-5 is cytokine that could control

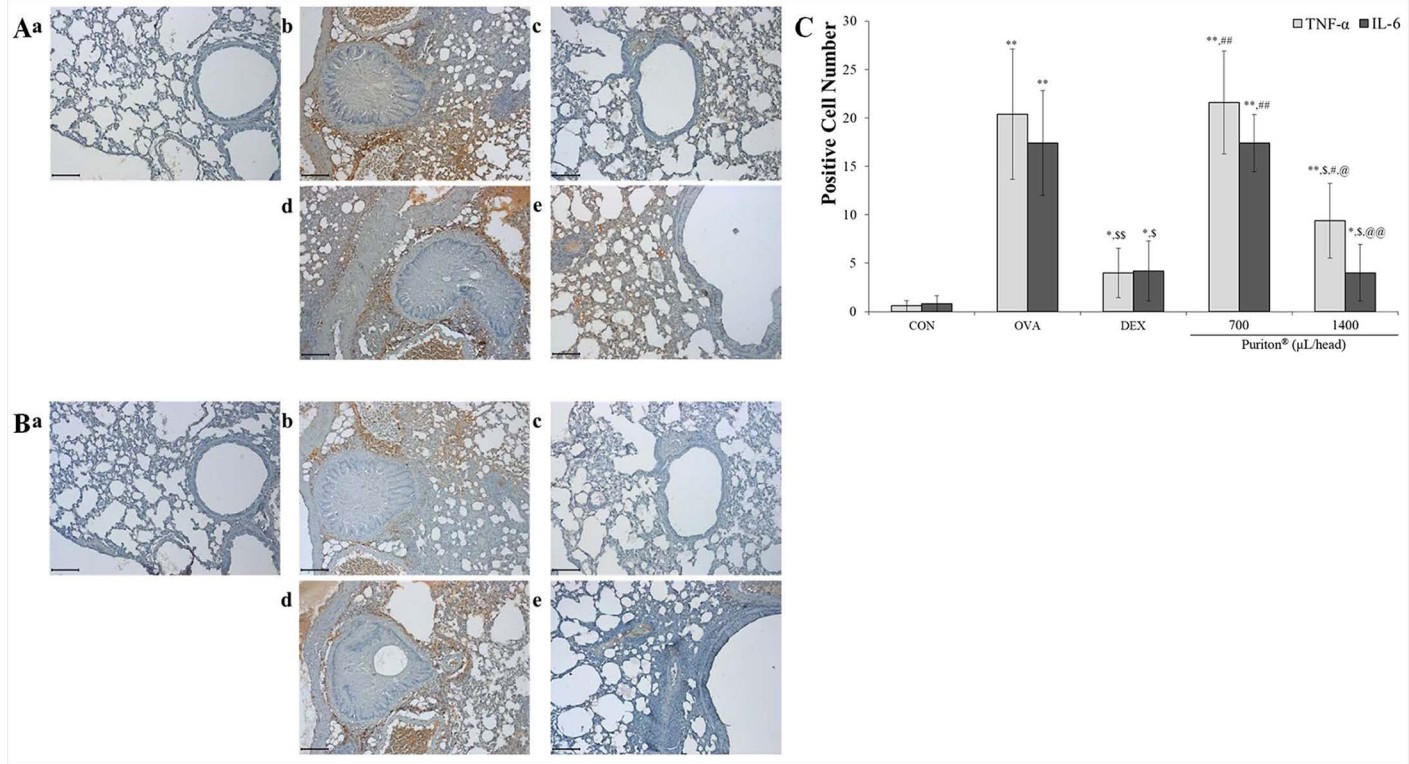

**Fig 5. Puriton controlled the expression of inflammation-related cytokines such as TNF- α and IL-6 and product, PGE₂.** (A) Puriton dose-dependently suppressed the expression of TNF-α. (B) Puriton completely controlled the expression level of IL-6 protein in the lung. (C) Puriton suppressed PGE₂ expression similar to that in DEX. a. CON, b. OVA, c. DEX, d. Puriton 700 μL/head, e. Puriton 1400 μl/head. Scale bar, 100 μm. Magnification, ×200. (D) The summary of anti-expression effects by Puriton on TNF-α, IL-6, and PGE₂. The results were described by mean ± SD. * $p < 0.05$ vs. CON, ** $p < 0.001$ vs. CON, $ $p < 0.05$ vs. OVA, $$ $p < 0.001$ vs. OVA, # $p < 0.05$ vs. DEX, ## $p < 0.001$ vs. DEX, @ $p < 0.05$ vs. Puriton 700 μL/head, @@ $p < 0.001$ vs. Puriton 700 μL/head.

not only airway hyperreactivity but also the activations of B cell and eosinophil via IgE recruitment [34,36,37]. Form these results, we could suppose that Puriton blocked IgE recruitment, suppressed IL-5 release, inactivated B cell and basophil, and finally suppressed airway hyperreactivity. Puriton not only regulates immune balance through IL-5 regulation but also suppresses IL-6 and TNF-α levels.

The NF-κB/COX-2/PGE₂ pathway is a significant inflammatory pathway [17–19]. The activation of this pathway is induced by inflammatory cytokines such as IL-1β and TNF-α, and it plays a role in inducing inflammatory responses [20]. Arachidonic acid is metabolised to prostaglandins and leukotrienes by COX-2 and lipoxygenase, respectively, resulting in elevated levels of these mediators [18]. Treatment of the subject with Puriton resulted in the inhibition of the expression of p-NF-κB and COX-2 in the nucleus and cytoplasm (Fig 6A). Furthermore, treatment with Puriton reduced the amount of PGE₂ expression increased by OVA (Fig 6B and 6C).

Dexamethasone, which has been used as a symptom modifier against asthma, [21] is a corticosteroid anti-inflammatory drug that can block the conversion of arachidonic acid from the plasma membrane. As this step is an initia stage, it can affect almost steps in inflammation. However, as it has many adverse effects, such as growth retardation, osteoporosis, immunosuppression, glaucoma/cataracts, and psychiatric problems [23–25], dexamethasone should be used under the limited condition. In contrast, each compound in the composition of Puriton is an edible mineral, and it had been known to modulate immune balance-related to helper T cells and inflammation, as demonstrated in this study.

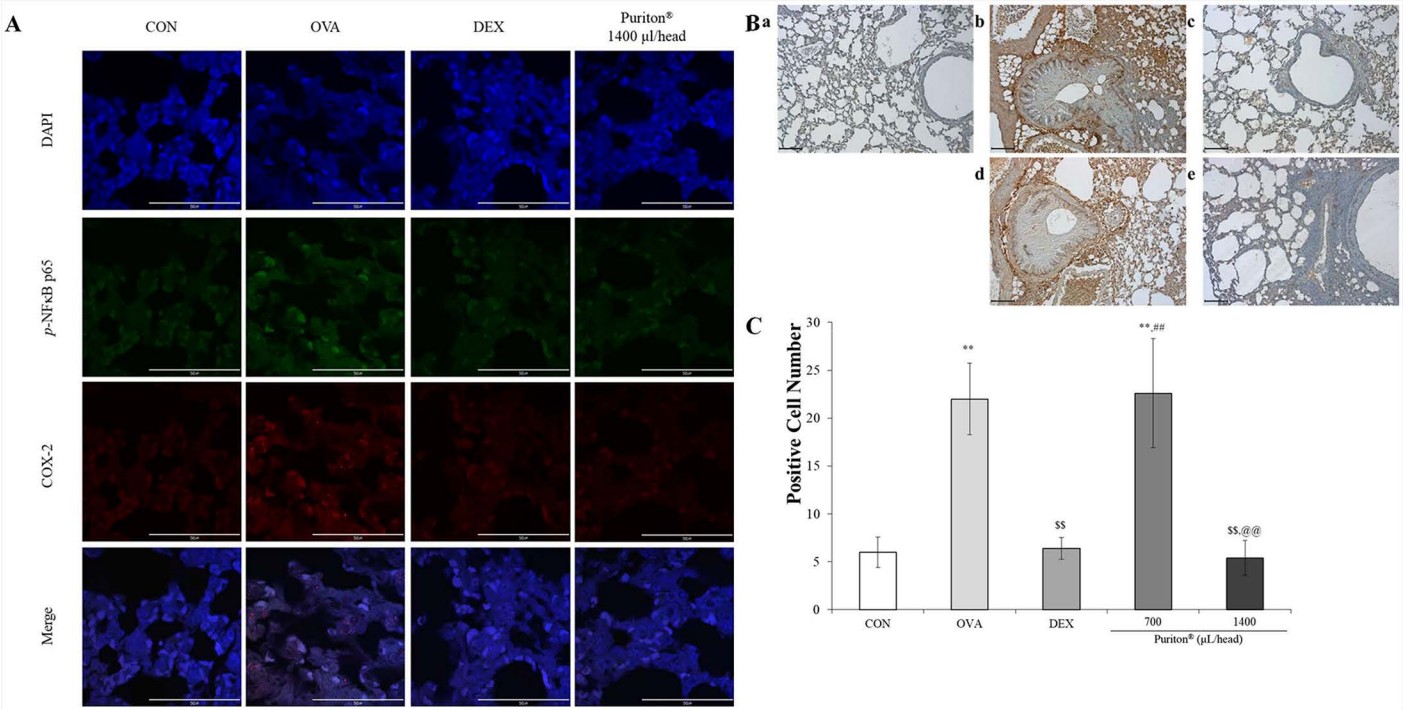

**Fig 6. Puriton inactivates NF- κB/ COX-2/PGE$_2$ pathway.** (A) Immunofluorescence revealed that Puriton led to a suppression of the expression of *p*-NF-κB (green fluorescence) and COX-2 (red fluorescence). The administration of 1400 μL/head/day of Puriton resulted in a comparable outcome to that observed in the DEX treatment group. The scale bar represents 50 μm, with magnification set at × 1000. (B) Immunohistochemistry for PGE$_2$ expression showed that Puriton treatment reduced the PGE$_2$ (dark brown) levels increased by OVA treatment. (C) Summary of the anti-expression effect of Puriton on PGE$_2$. Results are presented as mean ± SD. a, CON; b, OVA; c, DEX; d, 700 μL/head/day Puriton; e, 1400 μL/head/day Puriton. Scale bar, 50 μm; magnification, × 1000. Results are presented as mean ± SD. ** $p < 0.001$ vs. CON, $^{\$\$}$ $p < 0.001$ vs. OVA, $^{\#\#}$ $p < 0.001$ vs. DEX, $^{@@}$ $p < 0.001$ vs. Puriton 700 μL/head.

Puriton is a clear liquid containing a mixture of minerals in sterilized water with a pH of almost 7.0. It is used as an oral treatment and has an antibacterial effect [29]. Mineral are very important element not only in composing our bodies, such as bone and hemoglobin in blood and muscle, but also in regulating biological system, such as neurotransmission, heartbeat, and chemical reaction as a co-enzyme. From 2020 to April 2024, almost 16 thousand results related to keyword 'mineral therapy' could be found in PubMed. Most of them were associated with nutrient supplements and physiological or pathological calcification in the body, such as osteogenesis or diseases caused by calcium accumulation in some tissues. However, the results of biological efficacy about mineral treatments were also found. Minerals have been attributed to the regulation of the immune system and inflammation [44], control of wound healing [45], cancer therapy [46], anti-microbial effect [29,47], control of overuse pain and muscle tension syndrome control [48], and skin cosmetic therapy [49]. However, some articles addressed the toxicity of mineral, such as traditional Chinese medicine including mineral [50], talc inhalation [51], and carcinogenesis [52]. Although mineral have been used as nutrient and/or remedies worldwide, they have both biological efficacy and toxicity. Therefore, to apply minerals as biological regulators, it is necessary to prudently consider their toxicity at the same time they are used.

The anti-asthmatic effect was barely evident in the low-dose Puriton group; however, it was distinctly apparent in the high-dose Puriton group. This finding suggests a dose-dependent pattern in comparison to the effect observed in the low-dose Puriton group. However, to define a dose-dependent manner of Puriton treatment, further study is needed.

The administration of Puriton resulted in a reduction in the morphological changes observed in the pulmonary system, as well as a decrease in the number of leukocytes and neutrophils present in the BALF. Additionally, the levels of Th2 cell-related cytokines, including IgE, IL-4, IL-5, and IL-13, were diminished in the serum. The reduction in inflammatory factors such as TNF-α and IL-6, as well as the decreased expression of the important inflammatory pathway NF-κB/COX-2/PGE2, contributed to the effective control of asthma-related changes. As the Puriton dose for anti-asthmatic effect on mouse was evaluated at 1.4 mL/head, and the extrapolation factor for mouse is 13, the dose for humans is estimated at 233 mL/person. Based on these findings, Puriton can be considered a promising drug candidate for asthma management.

## Acknowledgments

Not applicable.

## Author contributions

**Conceptualization:** Dae-Hun Park.

**Data curation:** So-Hyeon Bok, Hae Eun Jang.

**Formal analysis:** So-Hyeon Bok.

**Investigation:** So-Hyeon Bok, Kwang-Ho Kim.

**Methodology:** So-Hyeon Bok, Hae Eun Jang.

**Project administration:** Dae-Hun Park.

**Resources:** Kwang-Ho Kim.

**Writing – original draft:** So-Hyeon Bok, Hae Eun Jang.

**Writing – review & editing:** Dae-Hun Park.

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
