## [Decision Letter · Decision Letter 0]

25 Feb 2025

PONE-D-25-03572Puriton® attenuates the asthma severity in ovalbumin-induced murine model via balancing Th1/Th2 and inhibiting inflammationPLOS ONE

Dear Dr. Park,

Thank you for submitting your manuscript to PLOS ONE. After careful consideration, we feel that it has merit but does not fully meet PLOS ONE’s publication criteria as it currently stands. Therefore, we invite you to submit a revised version of the manuscript that addresses the points raised during the review process.

We look forward to receiving your revised manuscript.

Kind regards,

Vinh Le Ba, PhD in Pharmaceutical Science

Academic Editor

PLOS ONE

Journal Requirements:

I have read the journal's policy and the authors of this manuscript have the following competing interests: Kwang-Ho Kim.

3. Please include a caption for figure 6.

4. Please upload a copy of Figure 6, to which you refer in your text on page 14. If the figure is no longer to be included as part of the submission please remove all reference to it within the text.

Reviewers' comments:

Reviewer's Responses to Questions

**Comments to the Author**

1. Is the manuscript technically sound, and do the data support the conclusions?

Reviewer #1: Yes

Reviewer #2: Yes

2. Has the statistical analysis been performed appropriately and rigorously? 

Reviewer #1: Yes

Reviewer #2: Yes

3. Have the authors made all data underlying the findings in their manuscript fully available?

Reviewer #1: Yes

Reviewer #2: Yes

4. Is the manuscript presented in an intelligible fashion and written in standard English?

Reviewer #1: No

Reviewer #2: Yes

5. Review Comments to the Author

Reviewer #1: The manuscript studied the attenuation effect of Puriton on allergic asthma in mice. Puriton may play a role in restoring the balance between Th1/Th2, improving airway remodeling, reducing mucus secretion and goblet cells hyperplasia, suppressing Th2-related cytokines as well as IgE production. It may be an effective and helpful treatment for allergic asthma. However, there are some problems, which must be solved before it is considered for publication.

The first major problem with this manuscript is that it is poorly grammatically written. I have listed some of them below.

Abstract:

- All abbreviations should be defined upon their first appearance.

Line 19: Change to “In 2019, 262 million asthma patients were estimated, with 455 thousand deaths caused by asthma”.

Line 21: “elder” replace by “elderly”.

Line 26: “Puriton” add “Treatment with Puriton”.

Introduction:

- From Introduction to Discussion, All abbreviations should be defined upon their first appearance and used consistently thereafter.

- Information about Puriton needs to be supplemented in INTRODUCTION.

Line 38, 39: grammar “Symptoms of asthma includes……which is …..[4] and airway…..” replaced by “Symptoms of asthma include……which are …..[4]. Airway…..”

Line 40: “was resulted” replace by “results” & “change” to “changes”

Line 42: “hyperplasia/apoptosis/fibrosis” change to “hyperplasia, apoptosis, and fibrosis”.

Line 48: “….of that will be over that” change to “….will be over that age”.

Line 49-51: long sentence.

Line 51: “…not only Th1 cell and Treg cells” replace by “…not only Th1 cells but also regulatory T (Treg) cells”.

Line 56: change to “ …Th1 and Th2 cells and Treg cell-related cytokines…..”

Line 59: delete “for”.

Line 60: change into “drugs are categorized into: anti-inflammatory……….”.

Line 62-63: “As most of all need inhalator (nebulizer) …..and that is therapeutic limit” rewrite.

Line 67: change to “………………gas exchange, fluid balance, and muscle contraction.”

Line 68: “And recently biological effects of them have induced…..” replace by “Recently biological effects of them have been induced……”. Do not start the sentence with “And”.

Line 70: “enhancing” replace by “enhancement”.

Materials and methods:

It is best described, however:

I think “Ethics statement” comes before “Animal experiment” .

Line 79: Remove “with”.

Line 81: “…as a positive one” replace by “…as appositive control”.

Line 82: change to “2 dosage groups of Puriton”.

Line 90: change to “…and in the afternoon, 5% ovalbumin was inhaled using nebulizer….”.

Line 115: change to “After fixation, the samples were dehydrated using graded …..”

Line 117: “sectioned by” replace by “sectioned at”.

Line 120-121: change to “……and Schiff stain (….) were conducted.”; “using with” replace by “using”.

Line 127: remove “with”.

Line 134: change to “prostaglandin E2 (PGE2)”

Results:

- Significant levels (P-values) must be added in your Result Section.

- Titles are so long.

- You must mention the increase or decrease in the results compared to which group.

Line 171: change to “As shown in Fig 3A. OVA………”.

Line 173: “seem” replace by “seemed or appeared”.

Line 176-178: change to “Although the immunohistochemical analysis suggested that the control effect of Puriton® on IFN-γ and IL-12p40 seemed effective based on quantitative evaluation, the effect of Puriton® did not reach statistical significance (Fig 3C)."”

Line182: add compared to what “The expression of IL-4 protein in OVA treatment group significantly increased compared to CON….”.

Line 187-190: IL-6!!!! You mean IL-13., replace.

Line 205: “by those in DEX” correct as “to those observed with DEX treatment”

Discussion:

Punctuation should be added.

Line 209: “…..meet allergen APCs introduce that to Th2 cell” replace by “…..meet allergen, APCs introduce it to Th2 cell”.

Line 211-215: change to “….(< 1 day), its high affinity for mast and basophil allows easy interaction with allergens. After a mast cell and/or basophil meets an allergen, leukotriene D4 and PGE2 are…… such as asthma, allergic rhinitis, and atopic dermatitis”.

Line 216-220: long sentence.

Line 220: change to “IL-4 released from…the release of IL-4, IL-5, and IL-13, promotes IgE production from….”.

Line 246: “such” change to “such as”.

Line 250: change to “….inflammation, as demonstrated in this study.”

Line 259: “Minerals attributed” replace by “Minerals have been attributed”.

Line 268-270: rewrite.

References:

It’s better to add recent references (2024 and/or 2025)

Figures:

- Any abbreviations used in the figures must be defined in figure captions upon their first appearance, then use the abbreviated form.

- Use arrows or stars to point out your results on the figures.

- Improve the resolution of figures.

Line 419, 428: “effects of” replace by “effects on”.

Reviewer #2: I found the article to be accepted with slight modifications, which are addressed in the suggestion, as NF-KB and signaling pathways are needed to say Puriton has an anti-inflammatory effect to treat asthma.

6. PLOS authors have the option to publish the peer review history of their article (what does this mean? ). If published, this will include your full peer review and any attached files.

**Do you want your identity to be public for this peer review?** For information about this choice, including consent withdrawal, please see our Privacy Policy .

Reviewer #1: No

Reviewer #2: **Yes: ** Dr. Vinita Pandey

---

## [Author Response · Author response to Decision Letter 1]

21 Mar 2025

Response to Journal Requirements

Ans) I would like to express my gratitude for the feedback I have received. Revisions have been made to the text in accordance with the comments received.

I have read the journal's policy and the authors of this manuscript have the following competing interests: Kwang-Ho Kim.

Ans) Thanks for your comment. I included it in my cover letter.

3. Please include a caption for figure 6.

Ans) Thanks for the comment. I've fixed it.

4. Please upload a copy of Figure 6, to which you refer in your text on page 14. If the figure is no longer to be included as part of the submission please remove all reference to it within the text.

Ans) Thanks for the comment. I included the information for Figure 6.

Response to editor’s comments

1. Justification for Group Division

Please provide a detailed justification for the division of 48 mice into five groups. Specify the criteria used for group allocation and describe the experimental rationale for this grouping strategy.

Ans) Thank you for your comments. A total of 40 animals were used, divided into five groups of eight, and this part has been revised.

2. Introduction Quality

The introduction is well-structured and effectively written, providing a comprehensive background and rationale for the study.

Ans) Thanks for the comment.

3. Rationale for Dividing Experimental Mice for BALF and Histological Analyses

Kindly clarify the rationale for dividing experimental mice into two subgroups: one for bronchoalveolar lavage fluid (BALF) analysis (n=4) and the other for histological examination (n=4). Given that these analyses could theoretically be conducted on the same group, explain why splitting the groups was deemed necessary.

Ans) Thanks for the comment. Considering the possibility that some damage to lung tissue may have occurred during the BALF collection process, we used 4 animals each and added them to manuscript.

4. Image Visibility in Immunohistochemistry Figures (Figure 2B)

Some images in immunohistochemistry figures (2B b and 2B d) are not clearly visible for proper visualization and analysis. Please ensure the quality and resolution of these images are sufficient for accurate interpretation.

Ans) Thanks for the comment, I fixed the resolution.

5. Th1/Th2 Cytokine Balance

The study demonstrates good connectivity regarding the balance between Th1 and Th2 cytokine release. However, the lack of signaling pathway analysis limits the mechanistic understanding of how Puriton attenuates asthma severity. It is recommended to address this gap without this anti-inflammatory effect of Puriton cannot be justificable.

Ans) Thanks for the comment. Added the NF-κB/COX-2/PGE2 pathway.

6. NF-κB Level Analysis

To strengthen the findings, it is suggested that the study should include an analysis of NF-κB levels. This would provide further insights into the underlying molecular mechanisms and support the proposed protective role of Puriton in asthma.

Ans) Thanks for the comment. I added p-NF-κB and COX-2, which are NF-κB pathways.

Response to reviewer #1’s comments

The manuscript studied the attenuation effect of Puriton on allergic asthma in mice. Puriton may play a role in restoring the balance between Th1/Th2, improving airway remodeling, reducing mucus secretion and goblet cells hyperplasia, suppressing Th2-related cytokines as well as IgE production. It may be an effective and helpful treatment for allergic asthma. However, there are some problems, which must be solved before it is considered for publication.

The first major problem with this manuscript is that it is poorly grammatically written. I have listed some of them below.

Abstract:

- All abbreviations should be defined upon their first appearance.

Ans) Thanks for the comment. I've fixed it.

Line 19: Change to “In 2019, 262 million asthma patients were estimated, with 455 thousand deaths caused by asthma”.

Ans) Thank you so much and we amended the sentence according to the comment.

Line 21: “elder” replace by “elderly”.

Ans) Thank you so much and we amended the sentence according to the comment.

Line 26: “Puriton” add “Treatment with Puriton”.

Ans) Thank you so much and we amended the sentence according to the comment.

Introduction:

- From Introduction to Discussion, All abbreviations should be defined upon their first appearance and used consistently thereafter.

Ans) Thanks for the comment. I've fixed it.

- Information about Puriton needs to be supplemented in INTRODUCTION.

Ans) Thank you for your comment. I have added the information of Puriton according to your comment to the main text.

Line 38, 39: grammar “Symptoms of asthma includes……which is …..[4] and airway…..” replaced by “Symptoms of asthma include……which are …..[4]. Airway…..”

Ans) Thank you so much and we amended the sentence according to the comment.

Line 40: “was resulted” replace by “results” & “change” to “changes”

Ans) Thank you so much and we amended the sentence according to the comment.

Line 42: “hyperplasia/apoptosis/fibrosis” change to “hyperplasia, apoptosis, and fibrosis”.

Ans) Thank you so much and we amended the sentence according to the comment.

Line 48: “….of that will be over that” change to “….will be over that age”.

Ans) Thank you so much and we amended the sentence according to the comment.

Line 49-51: long sentence.

Ans) Thanks for the comment. I've fixed it.

Line 51: “…not only Th1 cell and Treg cells” replace by “…not only Th1 cells but also regulatory T (Treg) cells”.

Ans) Thanks for the comment. When I amended above request (Line 49-51: long sentence) I've fixed it .

Line 56: change to “ …Th1 and Th2 cells and Treg cell-related cytokines…..”

Ans) Thank you so much and we amended the sentence according to the comment.

Line 59: delete “for”.

Ans) Thank you so much and we amended the sentence according to the comment.

Line 60: change into “drugs are categorized into: anti-inflammatory……….”.

Ans) Thank you so much and we amended the sentence according to the comment.

Line 62-63: “As most of all need inhalator (nebulizer) …..and that is therapeutic limit” rewrite.

Ans) Thanks for the comment. I've fixed it.

Line 67: change to “………………gas exchange, fluid balance, and muscle contraction.”

Ans) Thank you so much and we amended the sentence according to the comment.

Line 68: “And recently biological effects of them have induced…..” replace by “Recently biological effects of them have been induced……”. Do not start the sentence with “And”.

Ans) Thank you so much and we amended the sentence according to the comment.

Line 70: “enhancing” replace by “enhancement”.

Ans) Thank you so much and we amended the sentence according to the comment.

Materials and methods:

It is best described, however:

I think “Ethics statement” comes before “Animal experiment” .

Ans) Thanks for the comment. I've fixed it.

Line 79: Remove “with”.

Ans) Thank you so much and we amended the sentence according to the comment.

Line 81: “…as a positive one” replace by “…as appositive control”.

Ans) Thank you so much and we amended the sentence according to the comment.

Line 82: change to “2 dosage groups of Puriton”.

Ans) Thank you so much and we amended the sentence according to the comment.

Line 90: change to “…and in the afternoon, 5% ovalbumin was inhaled using nebulizer….”.

Ans) Thank you so much and we amended the sentence according to the comment.

Line 115: change to “After fixation, the samples were dehydrated using graded …..”

Ans) Thank you so much and we amended the sentence according to the comment.

Line 117: “sectioned by” replace by “sectioned at”.

Ans) Thank you so much and we amended the sentence according to the comment.

Line 120-121: change to “……and Schiff stain (….) were conducted.”; “using with” replace by “using”.

Ans) Thank you so much and we amended the sentence according to the comment.

Line 127: remove “with”.

Ans) Thank you so much and we amended the sentence according to the comment.

Line 134: change to “prostaglandin E2 (PGE2)”

Ans) Thanks for the comment. I've fixed it.

Results:

- Significant levels (P-values) must be added in your Result Section.

Ans) Thanks for the comment. I've added it to the text.

- Titles are so long.

Ans) Thanks for the comment. I've fixed it.

- You must mention the increase or decrease in the results compared to which group.

Ans) Thanks for the comment. I've fixed it.

Line 171: change to “As shown in Fig 3A. OVA………”.

Ans) Thank you so much and we amended the sentence according to the comment.

Line 173: “seem” replace by “seemed or appeared”.

Ans) Thank you so much and we amended the sentence according to the comment.

Line 176-178: change to “Although the immunohistochemical analysis suggested that the control effect of Puriton® on IFN-γ and IL-12p40 seemed effective based on quantitative evaluation, the effect of Puriton® did not reach statistical significance (Fig 3C)."”

Ans) Thank you so much and we amended the sentence according to the comment.

Line182: add compared to what “The expression of IL-4 protein in OVA treatment group significantly increased compared to CON….”.

Ans) Thank you so much and we amended the sentence according to the comment.

Line 187-190: IL-6!!!! You mean IL-13., replace.

Ans) Thank you so much for the comment. I've fixed it.

Line 205: “by those in DEX” correct as “to those observed with DEX treatment”

Ans) Thanks for the comment. I've fixed it.

Discussion:

Punctuation should be added.

Ans) Thanks for the comment. I've fixed it.

Line 209: “…..meet allergen APCs introduce that to Th2 cell” replace by “…..meet allergen, APCs introduce it to Th2 cell”.

Ans) Thank you so much and we amended the sentence according to the comment.

Line 211-215: change to “….(< 1 day), its high affinity for mast and basophil allows easy interaction with allergens. After a mast cell and/or basophil meets an allergen, leukotriene D4 and PGE2 are…… such as asthma, allergic rhinitis, and atopic dermatitis”.

Ans) Thank you so much and we amended the sentence according to the comment.

Line 216-220: long sentence.

Ans) Thanks for the comment. I've fixed it.

Line 220: change to “IL-4 released from…the release of IL-4, IL-5, and IL-13, promotes IgE production from….”.

Ans) Thank you so much and we amended the sentence according to the comment.

Line 246: “such” change to “such as”.

Ans) Thank you so much and we amended the sentence according to the comment.

Line 250: change to “….inflammation, as demonstrated in this study.”

Ans) Thank you so much and we amended the sentence according to the comment.

Line 259: “Minerals attributed” replace by “Minerals have been attributed”.

Ans) Thank you so much and we amended the sentence according to the comment.

Line 268-270: rewrite.

Ans) Thanks for the comment. I've fixed it.

References:

It’s better to add recent references (2024 and/or 2025)

Ans) We are grateful for the comments provided by the reviewer. Regarding the request for an up-to-date reference, it was decided that the current references would be maintained, as the existing literature provides a significant foundation for the topic.

Figures:

- Any abbreviations used in the figures must be defined in figure captions upon their first appearance, then use the abbreviated form.

Ans) Thanks for the comment. I've fixed it.

- Use arrows or stars to point out your results on the figures.

Ans) Thanks for the comment. I've fixed it.

- Improve the resolution of figures.

Ans) Thanks for the comment. I've fixed it.

Line 419, 428: “effects of” replace by “effects on”.

Ans) Thank you so much and we amended the sentence according to the comment.

Response to reviewer #2’s comments

I found the article to be accepted with slight modifications, which are addressed in the suggestion, as NF-KB and signaling pathways are needed to say Puriton has an anti-inflammatory effect to treat asthma.

Ans) Thank you so much for the comment and the NF-�B/COX-2 pathway was added.

---

## [Editor Report · Decision Letter 1]

28 Mar 2025

Puriton® attenuates the asthma severity in ovalbumin-induced murine model via balancing Th1/Th2 and inhibiting inflammation

PONE-D-25-03572R1

Dear Dr. Park,

We’re pleased to inform you that your manuscript has been judged scientifically suitable for publication and will be formally accepted for publication once it meets all outstanding technical requirements.

Kind regards,

Vinh Le Ba, PhD in Pharmaceutical Science

Academic Editor

PLOS ONE

Additional Editor Comments (optional): You can change the author information based on your comments. 
---

## [Editor Report · Acceptance letter]

PONE-D-25-03572R1

PLOS ONE

Dear Dr. Park,

I'm pleased to inform you that your manuscript has been deemed suitable for publication in PLOS ONE. Congratulations! Your manuscript is now being handed over to our production team.

Kind regards,

on behalf of

Dr. Vinh Le Ba

Academic Editor

PLOS ONE